# A Wearable Sensor Node for Measuring Air Quality Through Citizen Science Approach: Insights from the SOCIO-BEE Project

**DOI:** 10.3390/s25123739

**Published:** 2025-06-15

**Authors:** Nicole Morresi, Maite Puerta-Beldarrain, Diego López-de-Ipiña, Alex Barco, Oihane Gómez-Carmona, Carlos López-Gomollon, Diego Casado-Mansilla, Maria Kotzagianni, Sara Casaccia, Sergi Udina, Gian Marco Revel

**Affiliations:** 1Department of Industrial Engineering and Mathematical Sciences, Polytechnic University of Marche, 60131 Ancona, Italy; s.casaccia@univpm.it (S.C.); gm.revel@univpm.it (G.M.R.); 2DeustoTech, University of Deusto, 48007 Bilbao, Spain; mpuerta004@deusto.es (M.P.-B.); dipina@deusto.es (D.L.-d.-I.); a.barco@deusto.es (A.B.); oihane.gomezc@deusto.es (O.G.-C.); dcasado@deusto.es (D.C.-M.); 3Ayuntamiento de Zaragoza, 50003 Zaragoza, Spain; clopezg@zaragoza.es; 4Municipality of Amaroussion, 15100 Athens, Greece; mkotzagianni@maroussi.gr; 5Data Science Team, Bettair Cities SL, 08907 Barcelona, Spain; sudina@bettaircities.com

**Keywords:** citizen science, air quality, wearable devices

## Abstract

Air pollution is a major environmental and public health challenge, especially in urban areas where fine-grained air quality data are essential to effective interventions. Traditional monitoring networks, while accurate, often lack spatial resolution and public engagement. This study presents a novel wearable wireless sensor node (WSN) that was developed within the Horizon Europe SOCIO-BEE project to support air quality monitoring through citizen science (CS). The low-cost, body-mounted WSN measures NO_2_, O_3_, and PM_2.5_. Three pilot campaigns were conducted in Ancona (Italy), Maroussi (Greece), and Zaragoza (Spain), and involved diverse user groups—seniors, commuters, and students, respectively. PM_2.5_ sensor data were validated through two approaches: direct comparison with reference stations and spatial clustering analysis using K-means. The results show strong correlation with official PM_2.5_ data (R^2^ = 0.75), with an average absolute error of 0.54 µg/m^3^ and a statistical confidence interval of ±3.3 µg/m^3^. In Maroussi and Zaragoza, where no reference stations were available, the clustering approach yielded low intra-cluster coefficients of variation (CV = 0.50 ± 0.40 in Maroussi, CV = 0.28 ± 0.30 in Zaragoza), indicating that the measurements had high internal consistency and spatial homogeneity. Beyond technical validation, user engagement and perceptions were evaluated through pre-/post-campaign surveys. Across all pilots, over 70% of participants reported satisfaction with the system’s usability and inclusiveness. The findings demonstrate that wearable low-cost sensors, when supported by a structured engagement and data validation framework, can provide reliable, actionable air quality data, empowering citizens and informing evidence-based environmental policy.

## 1. Introduction

One of the most pressing environmental challenges faced by citizens, especially in densely populated urban areas, is air pollution. Rising pollution levels not only threaten public health but also have severe environmental consequences (e.g., climate change), and this makes air quality a critical concern in the European Union (EU) and at the global level. In the EU alone, poor air quality is linked to approximately 400,000 premature deaths each year. Additionally, it is a major factor contributing to biodiversity loss, which further exacerbates ecological imbalances [1]. The COVID-19 pandemic, along with its relationship to climate change, further highlighted the need for a healthier planet. During the pandemic, lockdown policies drastically altered energy demand patterns, resulting in a significant drop in CO_2_ emissions (up to 17%) due to reduced surface transportation [2]. This reduction underscores the potential impact of both policy measures and individual and collective societal actions in mitigating air pollution. Measuring the air quality in urban settings is the most fundamental step in raising awareness, in a sustainable and informed way, of the necessity of reducing CO_2_ emissions and air pollution. Although most cities are equipped with standardized reference station for measuring air quality, the capillarity of these networks is reduced due to the high costs that are involved. To face this challenge, low-cost sensors (LCS) for measuring air quality are being used more and more frequently, and are typically combined with social approaches that aims to increase responsibility and citizen engagement. In fact, citizen science (CS) has emerged as a scalable strategy to complements conventional air quality monitoring networks through the deployment of LCS technologies operated by non-expert users in order to increase the spatial resolution of air quality measurements [3,4]. When properly designed, CS measurement campaigns not only support data-driven research but also promote awareness, community engagement, and participatory environmental governance. Many CS initiatives have emerged from community efforts aimed at resolving issues that directly impact local communities. This collaboration expands the scope and reaches of scientific endeavour, fosters community engagement, enhances scientific literacy, and promotes pro-social and pro-environmental behaviours [5,6].

In the air quality context, CS has shown potential in extending monitoring coverage and capturing localized pollution dynamics. However, its integration into scientific and policy frameworks remains constrained by several technical and operational challenges [7]. A key technical limitation lies in the quality and reliability of the data obtained through LCSs, as varying participant expertise may lead to inconsistencies in data collection, which may be affected by factors such as sensor drift, cross-sensitivity to interfering compounds, and improper usage conditions [3,8]. The degree of digitalization, sensor handling, and data recording practices across diverse participant groups can introduce inconsistencies, which raises concerns about the representativeness and accuracy of the collected data. In addition, social challenges such as ensuring sustained user participation over time and translating citizen-generated data into actionable evidence for public policy remain substantial. LCSs should be designed as intuitive sensor systems that can be incorporated into robust digital platforms with transparent feedback mechanisms that ensure engagement without imposing cognitive or technological barriers. LCSs must be designed to be physically unobtrusive, low-power, and user-friendly, while core features should be embedded for data validation, such as built-in diagnostics, remote logging, or adaptive sampling strategies [9,10,11]. Within this framework, the SOCIO-BEE (Wearables and Drones City Socio-Environmental Observations and BEhavioral ChangE) project addresses the challenges of citizen science-based air quality monitoring by introducing a methodology for conducting CS campaigns that is supported by a wearable wireless sensor node (WSN) capable of measuring NO_2_, O_3_, and PM_2.5_. This device is integrated into a structured engagement framework to actuate measurement campaigns and to ensure that the data collected by citizens are of sufficient quality to support both scientific analysis and policy development. The WSN is employed to obtain data through campaigns implemented in Zaragoza (Spain), Maroussi (Greece), and Ancona (Italy) with different societal groups. Data collected through the WSN by citizens during campaigns are analyzed and validated with a two-fold approach: (i) a direct comparison with official reference station data and (ii) a clustering-based validation method that employs K-Means algorithms to group daily measurements and assess intra-cluster coherence and variability, particularly in locations where reference data are not available. This research aims to present the technical innovation of SOCIO-BEE, which is supported by a multi-domain integration of citizens across all stages of the scientific process, including data collection, analysis, and interpretation; thus, this study enhances public awareness, fosters behavioral engagement, and reinforces the role of bottom-up contributions in evidence-based policymaking.

The structure of this paper is organized as follows. Section 2 reviews related work on air quality sensors adopted in CS initiatives with the aim of democratizing CS and improving the validity of crowdsourced air quality data. Section 3 provides an overview of the SOCIO-BEE methodology, the technical details of the WSN, the design of the measurement campaigns across pilot cities, and the data analysis to assess the data validity. Section 4 presents the results of the SOCIO-BEE project. Finally, Section 5 summarizes the conclusions of this paper and discusses the lessons learned throughout the project.

## 2. State of the Art

In line with the growing necessity of expanding air quality measurements in cities and with the adoption of CS, the past two decades have seen a notable increase in air quality monitoring sensors and related initiatives. Various projects have employed a wide range of sensor technologies and methodological frameworks to assess the feasibility, reliability, and scalability of low-cost sensing solutions [12,13]. As an example, the Luftdaten project [14] has notably empowered citizens to participate in air pollution data collection using accessible LCSs. This initiative has highlighted the importance of accessible technology in enabling widespread participation and has provided a framework for community-driven environmental monitoring. The OpenSense project [15] utilized mobile sensors and machine learning to analyze urban air quality. Similarly, a study conducted in Nairobi [16] analyzed the impact of the construction of a new highway. It involved distributing optical sensors (AlphaSense OPC-N2) among 15 households within 150 m of each other that were grouped into three clusters. Data collected over 48 h indicated air quality levels substantially above WHO-recommended thresholds, with a recorded average concentration of 28.7 μg/m^3^, in contrast to the WHO limits of 5 μg/m^3^ (annual) and 15 μg/m^3^ (daily). Another study [17] aimed to demonstrate the effectiveness of high-resolution sensor networks, compare collected data with a reference station, analyze air pollutants, and assess personal exposure. Using Sensirion SPS30 sensors integrated into the BMD-340 module with Bluetooth connectivity, approximately 40 sensors were distributed to citizens with the aim of capturing particulate matter (PM), thermo-hygrometric parameters, and gases such as CO_2_ and NO_2_. Each sensors costs approximately $250 at low production volumes, and less than $100 at large scale production. The measurements from the low-cost sensors closely matched those of the reference station, with PM_2.5_ averages of 3 μg/m^3^ (sensors) and 4.5 μg/m^3^ (reference). Bousiotis et al. proposed a hyperlocal, mobile source apportionment methodology in Birmingham that used a backpack-based sensor kit carried by citizen volunteers, integrating low-cost instruments such as the Alphasense OPC-N3. The study not only demonstrated the technical feasibility of attributing PM sources using positive matrix factorization (PMF) but also emphasized how mobile sensor-based CS can enhance the spatial resolution in urban pollution studies, especially when it is combined with structured calibration efforts against reference stations [18,19]. The literature also showed how the integration of low-cost CO_2_ and PM sensors into a public building (via the Guildford Living Lab) enabled real-time air quality visualization and public engagement. The study highlighted how short indoor events could elevate the CO_2_ concentration above 1000 ppm, and how human presence increased the PM10 via resuspension effects.

The presented literature raises an important question regarding the quality of data collected by LCSs, as these devices typically exhibit greater uncertainty compared to official monitoring stations. Indeed, due to their susceptibility to anomalies and external influences, especially the relative humidity and temperature, it remains challenging to perform thorough and accurate calibration of these sensors. Nevertheless, the ambition of SOCIO-BEE has been, precisely, to provide a LCS (the price tag for the device designed in the project was EUR 800) that is capable of delivering measurements that are sufficiently reliable, as demonstrated through direct comparisons against ground truth provided by reference stations, and to go further by testing these devices in real test-beds.

The SOCIO-BEE initiative and experimental measurement campaigns aimed to go further beyond traditional CS frameworks by integrating advanced wearable technologies and real-time data acquisition. This research applies clustering techniques to analyze the datasets collected through CS campaigns, comparing the clustered results with measurements obtained from reference stations. This approach allows researchers to evaluate whether data provided by citizen science initiatives are sufficient to accurately characterize air quality conditions, or if further data collection and methodological refinements are necessary. Several studies have explored the application of K-means clustering to spatially distributed air quality data, with a particular focus on particulate matter concentrations. For instance, some works [14,15] have applied K-means to geo-referenced PM measurements collected across urban environments to identify localized pollution patterns. Studies have also used spatial clustering to define air quality zones and assess the spatial behavior of PM_2.5_ and PM_10_ in relation to official monitoring infrastructure. These works incorporate geographic coordinates as clustering features, which allows for the derivation of spatial centroids and distance-based assessments, a methodology that is closely aligned with the approach followed in this research work.

## 3. SOCIO-BEE Methodology

This section presents the methodology developed by the SOCIO-BEE project to conduct CS experiments across different case studies and test the validity and quality of the data obtained thought the WSN, as well as the methodology used to measure the impact of the experiments.

### 3.1. The Bee Metaphor: The Social Structure of the Community of Practices

Drawing inspiration from nature, SOCIO-BEE, in the design and execution of its measurement campaigns, uses the metaphor of bee colonies to develop effective behavioral and engagement strategies for a diverse range of stakeholders. Figure 1 illustrates the roles that hive members, representing a community of practice, can assume within SOCIO-BEE’s approach, and also highlights the core features that the SOCIO-BEE technical platform must exhibit to enable members to oversee and actively participate in CS campaigns. It is important to note that the entire process undertaken by hive members is the result of a collaborative co-creation effort involving multiple stakeholders. The roles are inspired by the organization of a bee colony, and are intended to facilitate collaborative citizen engagement. The Beekeeper is responsible for initiating and coordinating the campaign, forming the team (or “hive”), and ensuring smooth collaboration among participants. The Queen Bee designs the campaign by defining key parameters, such as the location, measurement frequency, and duration, using the SOCIO-BEE web platform and templates. The Queen Bee also oversees the distribution of WSNs and manages the overall execution, including data collection, monitoring, and dissemination of the results. Worker Bees, the main data collectors, are notified via the SOCIO-BEE mobile app and guided to measurement locations by the MVE. They contribute air quality data, track campaign progress, and access feedback tools and heat maps during and after the campaign. Finally, Drone Bees act as ambassadors, sharing insights generated by the campaign and helping to raise awareness among institutions and the public.

### 3.2. Wearable Sensor Node

The WSN (Figure 2) is a low-cost, wearable device designed by Bettair® Cities for making air quality measurements. The measurement device integrates two slots for toxic gas sensors, a PM sensor, and an environmental sensor for the internal temperature and outdoor relative humidity. Gas sensors are commercial electrochemical sensors (4 series form factor) designed to detect NO_2_ and O_3_. These sensors require precision analog electronics, including a stable potentiostat and a low-noise transimpedance amplifier, to convert nanoampere-level signals into millivolt readings. The PM sensor measures PM_2.5_ concentrations. The WSN can operate autonomously on battery power, but it can also be connected to an external power source or power bank for continuous or prolonged monitoring. The WSN communicates with mobile phones via Bluetooth Low Energy (BLE), which enables the real-time transmission of data to the related cloud platform along with the phone’s GPS coordinates. These sensors can also connect directly to the cloud via Wi-Fi, but GPS data are not included in this mode. For pre-deployment calibration, Bettair® Cities performed factory calibration using proprietary, high-performance technology validated in field conditions. The initial accuracy of the toxic gas sensors aligned with the Class 2 specifications under the TS17660 standard and was expected to remain within at least Class 3 throughout the measurement period. The WSNs are housed in compact enclosures, as illustrated in Figure 2. The internal sampling rate is configurable, with a typical setting of one measurement every 5 s. However, the data are only considered reliable after a minimum of one minute of static acquisition at a fixed location. Regarding the potential influence of human metabolism on the sensors’ readings, it is important to note that, while indoor environments are more susceptible to localized effects caused by human occupancy, limited dispersion, and the resuspension of particles, outdoor environment and air pollution is predominantly driven by industrial activities and vehicular emissions. Consequently, indoor measurements are more vulnerable to human-induced interference and require careful consideration during calibration and sensor deployment. However, in the context of the present study, the SOCIO-BEE pilot measurement campaigns were conducted primarily in outdoor environments, where human emissions are significantly diluted and considered negligible relative to ambient pollutant sources [17,18,19]. Moreover, the WSNs were predominantly used in a stationary configuration, which allowed the sensors sufficient time to stabilize before each measurement. This strategy helps to reduce transient fluctuations and minimizes uncertainties associated with body heat, respiration, and movement. The supporting literature consistently indicates that indoor environments exhibit elevated particulate concentrations that are linked to human presence, while outdoor measurements are more influenced by environmental variability and are less affected by direct human metabolic outputs.

The WSN was developed together with the SOCIO-BEE infrastructure to facilitate CS initiatives that are focused on improving air quality and comprises the following additional components:-AcadeMe platform: This is the main entry point to the SOCIO-BEE system. It manages participants’ data, creates analytics, and connects with other SOCIO-BEE components (e.g., WSN). The AcadeMe platform is further divided into two core subcomponents:Mobile App: This application (see Figure 3) enables the participant to pair the WSN and consequently upload data collected by the WSN;Web app: Complementing the mobile app, the web app (see Figure 4) provides users with interactive visualizations of the WSN data, project updates, and opportunities to participate in discussions and forums with fellow citizen scientists and project stakeholders. The web app is important because a key feature is the possibility to define key areas to enable CS measurement campaigns through the use of WSNs;-Micro-Volunteering Engine (MVE) [20]: This microservice plays a key role in organizing citizen engagement and ensuring the collection of valid data within SOCIO-BEE’s CS campaigns. It provides users with tailored recommendations for specific locations where air quality measurements are needed [21]. The MVE optimizes volunteer efforts by guiding users to take measurements in the most critical areas, ensuring comprehensive and effective data coverage. Powered by diverse spatial computing strategies, it seeks to encounter the best balance between a given campaign’s measurement objectives, e.g., five measurements per hour per cell in the area covered, and the users’ behavior patterns and needs, e.g., some users do not mind walking for longer periods whilst others want to stick to taking measurements in their immediate surroundings.

### 3.3. Measurement Campaign

Prior to describing how the WSN were distributed to involved citizens, it is important to underline the fundamental concept pinpointed by SOCIO-BEE, which is based on the involvement of citizens in the measurement process. The experiments were designed to generate crowdsourced air quality data that can support actions to reduce pollution which are based on insights from data analysis. To evaluate the real-world applicability and the operational performance of the WSNs and the methodology, the system was deployed across three European cities, Ancona, Maroussi, and Zaragoza, through a structured execution. The implementation spanned three phases: (1) a pre-pilot phase focused on device testing, user training, and system refinement; (2) a pilot phase involving continuous field deployment of the wireless sensor nodes (WSNs), real-time data collection, and user engagement under varying urban conditions; and (3) a post-pilot phase aimed at evaluating sensor performance, collecting user feedback, and validating the data quality. This paper is focused on phase (2) and (3) of the pilot application.

In each pilot site, recruited participants were provided with dedicated training sessions to ensure proper use of the WSNs. These sessions included both technical and practical guidance on how to operate the devices, collect data reliably, and interact with the supporting digital tools (e.g., mobile app and web platform).

As part of the training, the users were informed about specific operational constraints identified during the preliminary testing and calibration phases to prevent the WSNs from negatively affecting data quality and ensuring proper device performance in real-world conditions. One of the first issues encountered was the tendency of the devices to overheat when charged in warm environments. The participants were advised to avoid placing the WSNs in direct sunlight or high-temperature conditions during charging, as excessive internal heat could affect the sensor’s stability or even damage its components. In addition, it was noted that sensor readings taken immediately after powering on the devices tended to be unstable. To address this, users were instructed to turn on the WSNs well in advance, ideally a few hours before initiating a measurement session, to allow the sensors to reach thermal and operational stability. It is important to note that, when worn on the human body, wearable particulate matter sensors may be influenced by proximity effects such as thermal plumes, resuspension due to movement, and emissions from respiration, as highlighted in the recent literature [19,20]. These factors were considered in user training and data interpretation to minimize measurement artifacts. To minimize such interferences, the participants were instructed to remain still during the one-minute measurement intervals and to wear the device attached either to a belt or a backpack to ensure consistent positioning and airflow exposure.

By communicating these technical constraints clearly, the SOCIO-BEE team aimed to enhance participant awareness of potential measurement artifacts and ensure more consistent and meaningful data. Data used for the analysis were collected from citizens during the pilot phase, which involved continuous field deployment of the WSNs.

### 3.4. Pilot Description

Ancona, a coastal city in central Italy with notable air quality challenges due to traffic congestion and environmental conditions, was selected as a SOCIO-BEE pilot site. The campaign focused on assessing the exposure to PM_2.5_ among vulnerable populations, particularly seniors, who represent 26% of the local population, to evaluate the sensors’ performance under real-world conditions and encourage outdoor activity in low-exposure zones. The Ancona pilot lasted approximately three months. The participants were voluntarily recruited through local associations, and each user was given the device for about two weeks before returning it.

The second city was Maroussi, a densely trafficked district in the Athens metropolitan area that was selected as a SOCIO-BEE pilot site due to its high commuter flow and associated air pollution. The campaign focused on commuters as both contributors to and victims of poor air quality. WSNs were deployed among a diverse group of participants, including residents, students, employees, and public transport users, to monitor their exposure to air pollutants in transit and workplace environments. The pilot enabled evaluation of the sensors’ performance across varied urban mobility patterns and supported inclusive engagement in real-world measurement scenarios. Similarly, the Maroussi pilot recruited citizens for a period of three months.

Zaragoza, a major urban center in northeastern Spain, hosted a SOCIO-BEE pilot that targeted students aged from 11 to 16 years old. The campaign aimed to assess the usability and educational impact of wearable low-cost air quality sensors within school-based activities. The students collected, analyzed, and interpreted environmental WSN data, contributing to both personal exposure assessment and broader urban air quality mapping. The pilot also served to test and refine the SOCIO-BEE platform’s hardware and software tools in an educational context, leveraging Etopia’s digital innovation hub and its school outreach network. Also in this case, measurement campaigns were conducted in school during lessons.

### 3.5. Data Analysis

First of all, while the system includes electrochemical sensors for NO_2_ and O_3_, its use was deliberately limited in the analytical phase of the study and the analysis was focused on PM_2.5_ data. This decision was based on several operational constraints and scientific limitations identified during preliminary testing. To assess the reliability of the air quality data collected through the citizen-operated WSNs, two complementary data analysis approaches were applied, each adapted to the specific data availability of the pilot cities, which provided different data according to the population involved in the measurement process. Notably, all measurements during the pilot campaigns were conducted in uncontrolled real-world environments which differ significantly from the controlled settings used during initial calibration. This raised the need to evaluate whether such conditions negatively affect the accuracy and consistency of the data. As the literature currently lacks systematic studies assessing the reliability of wearable LCS data in real-world citizen science contexts, this work applies two distinct methods to explore this issue from multiple perspectives.

The first method involved a direct comparison between air quality measurements obtained from WSN devices and those from reference monitoring stations, and this analysis was conducted for the city of Ancona. This involved aligning temporal data collected from WSNs with those from the nearest official air quality reference stations. The reference instrument used for measuring PM_2.5_ was a BAM 1020 Beta Attenuation Mass Monitor (Figure 5). Key statistical metrics were computed, including the mean absolute error (MAE), root mean square error (MSE), mean bias error, and coefficient of determination (R^2^), to quantify the accuracy and consistency of the citizen-collected PM_2.5_ measurements.

The second analysis involved spatial clustering, and was applied to citizen-collected data points using K-means:The approach was different for the city of Ancona: for each identified cluster, the mean (*µ_ci_*) and standard deviation (*σ_ci_*) of the PM_2.5_ values were computed. These values were then compared to the daily average values (*µ_d_*) from the reference stations for the city of Ancona, since reference data were available. To evaluate the spatial distribution, the centroid in the GIS coordinates of each cluster was derived and its geographic distance to the reference station was measured. The relationship between this distance in meters from the reference station and the associated relative error (difference from the reference value in percentage) were analyzed to understand the influence of proximity on measurement reliability;For Maroussi and Zaragoza, where no official station data were available for comparison, the analysis focused on intra-cluster variability in the citizen-collected data. K-means clustering was applied to group measurements spatially, and for each cluster, the coefficient of variation (CV) was computed to evaluate the dispersion of values relative to the average value. This method allowed us to assess the reliability and homogeneity of the measurements within spatial clusters, and thus to obtain conclusions on data quality even in the absence of a formal reference system.

The variation within each cluster serves as an indicator of internal consistency: a low coefficient of variation implies homogeneity of the measurements and suggests that sensors in the same area captured similar exposure levels. In contrast, a high variation may indicate environmental heterogeneity, sensor discrepancies, or localized pollution events. Therefore, internal consistency metrics should become a proxy for data quality, as they allow researchers to assess whether citizen-collected measurements are sufficiently coherent to support meaningful interpretation.

### 3.6. Impact Assessment Instruments and KPIs for the Democratization Evaluation

The SOCIO-BEE project has developed and employed a comprehensive set of impact assessment instruments for the participants who are recruited and the effectiveness of its citizen science campaigns from a social perspective. Below are the key instruments used to measure SOCIO-BEE impact through the execution of its CS campaigns:Data Model Queries and Logging Mechanism: This tool tracks the usage of SOCIO-BEE’s resources by executing SQL queries over its data model to gather quantitative key performance indicators (KPIs). For example, in its data model, it records the number of participants collecting air quality data via wearable sensors, it records every measure performed by each user with each concrete device, and so on;Demographic and Activity Satisfaction Questionnaires: These questionnaires collect demographic information (age range, gender, educational level, and so on) and assess participant satisfaction with various pilot activities;Evaluation Questionnaires: Designed for hive members involved in CS experiments or receiving results—including worker bees and drone bees—this evaluation process consists of the PRE and POST SOCIO-BEE assessment questionnaires:The PRE SOCIO-BEE Citizen Science Activists Evaluation Questionnaire is completed before the campaign begins to establish a baseline, i.e., a departure point, regarding participants’ awareness of the citizen science approach and its potential;The POST SOCIO-BEE Citizen Science Activists Evaluation Questionnaire is completed after the campaign concludes to measure changes and learning experienced by participants across the CS campaign.

The above two questionnaire types (the sociodemographic and evaluation ones) assess critical factors such as socio-demographics, acceptance, interests, attitudes, satisfaction, accessibility, inclusiveness, awareness of air quality issues, usability, and behavioral change. Following an *ex-ante* (i.e., pre) and *ex-post* (i.e., post) evaluation approach [22], they enable the measurement of changes in those taking part in CS campaigns before and after they participate in CS campaigns.

## 4. Results

A summary analysis of the main characteristics of the campaigns conducted in the three pilot cities is displayed in Table 1 together with a visual representation of the positions in the three cities where CS campaigns were conducted (Figure 6). In the city of Ancona (Table 2), between 27 May and 27 June 2024, one campaign was conducted, with six different sub-campaigns, representing six different areas of the city (Figure 7); 10 recruited citizens were involved who collectively performed a total of 225 measurements. Over the course of just over a year, from 6 June 2023 to 28 June 2024 (measurements of 30 days in that period were collected), a total of 34 measurement campaigns were conducted in Maroussi (Table 3), with the active participation of 59 citizens. These campaigns involved 61 participants who collectively performed 1206 measurements, contributing valuable data on the air quality in the city (Figure 8).

In Zaragoza (Table 4), over the course of 29 days, from mid-May to the end of June 2024, a total of 13 measurement campaigns were conducted. These campaigns involved 79 participants who collectively performed 1206 measurements, contributing valuable data on the air quality in the city (Figure 9).

First, a summary analysis of the measurement campaigns that were conducted is presented. This includes relevant summary statistics that describe the overall trend of each campaign, including the number of users taking measurements, number of measurement points, and average PM_2.5_ concentration over the monitored period.

The Ancona seniors recorded lower, stable levels (residential areas), the Maroussi commuters showed traffic-linked peaks, and Zaragoza’s schoolchildren had extreme short-term spikes (e.g., Campaign 13). The differences reflect the measurement duration, localized sources, and participant behavior, underscoring the potential of citizen-collected data and methodological considerations for reliability. The single-day PM_2.5_ measurements in Zaragoza recorded an extreme outlier, likely due to transient pollution events near schools. The outlier in the data underscores the influence of localized, short-term environmental factors on air quality data. The campaigns highlight traffic-linked variability, with extremes such as Campaign 4 in Maroussi (21.3 ±13.3 µg/m^3^). The short-term campaigns (1–5 days) and diverse measurement points (10–139) captured transient exposure patterns near transit routes.

### 4.1. Comparison Between Measurements Obtained from WSN Devices Reference Stations

In this subsection, the data examined in the previous section for Ancona, where access to reference station data was possible, are compared to the reference monitoring stations to verify the validity of the data obtained through the CS campaigns.

Figure 10 presents the temporal evolution of the PM_2.5_ concentrations recorded by the official reference station in Ancona alongside the measurements collected during the CS campaigns, as well as the result of the regression analysis.

To quantify this relationship, a linear correlation analysis was conducted, the results of which are summarized in Table 5. The accuracy, expressed as the average of the residuals, is 0.54 µg/m^3^, and the precision (standard deviation of the residuals) is 2.2 µg/m^3^, suggesting that most CS measurements fall within a reasonable range of the reference data. Overall, the statistical confidence, with a coverage factor (K) of two, is 3.3 µg/m^3^ and the bias is 3.9 µg/m^3^.

### 4.2. Clustering Data Collected from the WSN

A cluster analysis was designed to identify discrepancies between the reference station and the citizen-collected campaign measurements. This analysis aims to categorize the daily data into distinct clusters using clustering methods. It serves a dual purpose: first, to determine whether each cluster collectively aligns with the trend observed at the reference station; and second, to evaluate the impact of the distances between the cluster centroids and the reference station’s location. Additionally, the variability in the daily collected data within each cluster is examined to further characterize the consistency and distribution of measurements.


**Ancona**


For the city of Ancona, the data were pre-processed and subsequently aggregated into hourly and daily datasets. The daily data were then input into the k-means algorithm, considering two configurations: cluster numbers of K = 2 and K = 3. An example of the clustering results is provided in Figure 11.

For each cluster, we computed the daily mean of the measurements and compared it with the daily mean from the reference station, calculating the associated relative error. The comparison between the cluster measurements and reference station data revealed a correlation coefficient (*r*) of 0.75 and 0.63 for clusters 0 and 1, respectively, for the case of K = 2. For K = 3, the *r* was 0.74, 0.63, and 0.68 for K = 0, 1, and 2, respectively. Despite these differences, the high correlation between cluster measurements and the reference station suggests that the data can be considered qualitatively reliable (Figure 12).

From a measurement discrepancy perspective, we analyzed the mean percentage errors for K = 2 and K = 3, computed against the data collected from the RSs (Table 6).

For the Ancona pilot, CS measurements taken farther from the reference station were compared to understand whether they remained meaningful. For this purpose, the GIS coordinates of each cluster centroid were considered and, for each measurement day, the distance between the centroid and the reference station was computed. Table 7 reports the correlation coefficient *r* between the relative error for each day, and the average distance from the RS.


**Maroussi**


As previously mentioned, in this analysis, a comparison among data collected from the WSNs was conducted, in order to analyze the variability in these data. As a first step, the daily clustering approach was applied using K = 2 and K = 3. For each clustering configuration, the mean PM_2.5_ concentration and the data dispersion (expressed as the standard deviation) were computed. The coefficient of variation (CV) was then calculated for each cluster, and the results were averaged across all days. Moreover, the analysis aimed to assess the internal consistency of the measurements and to determine whether closely spaced sensors tend to report similar values. A low CV would suggest homogeneous readings within a cluster, supporting the reliability of the data. Conversely, a high CV may indicate local anomalies, sensor instability, or contextual environmental differences, all of which may require further investigation or validation.

Figure 13 illustrates the distribution of the average PM_2.5_ concentrations for each cluster on each day, along with the associated dispersion, highlighting both the temporal and spatial variability. Table 8 summarizes the overall results, showing that, for K = 2, the standard deviation averaged 5.9 ± 6.9 µg/m^3^ with a mean CV of 0.50 ± 0.40, while for K = 3, slightly lower values were observed (5.8 ± 7.5 µg/m^3^ and CV = 0.5 ± 0.4). These results suggest that increasing the number of clusters may slightly reduce the internal dispersion, potentially improving the spatial resolution of the analysis.


**Zaragoza**


Similarly, the analysis was conducted in Zaragoza. Table 9 summarizes the overall results, showing that, for K = 2, the standard deviation averaged 2.3 ± 1.6 µg/m^3^ with a mean CV of 0.30 ± 0.30, while for K = 3, slightly lower values were observed (2.1 ± 1.7 µg/m^3^ and CV = 0.28 ± 0.30). Also, in this case, increasing the number of clusters reduced the internal dispersion, potentially improving the spatial resolution of the analysis. Figure 14 illustrates the distribution of the average PM_2.5_ concentrations for each cluster on each day, along with the associated dispersion, highlighting both the temporal and spatial variability.

### 4.3. Impact Assessment and KPIs

This section examines the impact of the SOCIO-BEE solution and measurement campaigns from multiple perspectives. It analyses age, gender, role, education level, digital proficiency, and employment status across the pilots. The data presented include all citizens who responded to PRE- or POST-campaign surveys, or both, in each city. Figure 8 illustrates the gender distribution of the participants. Most users chose to specify their gender, identifying as either male or female. Zaragoza and Maroussi showed a relatively balanced distribution between men and women, whereas Ancona had a higher proportion of male participants. Notice that in the case of Zaragoza, although measurements in the name of only 79 citizens were collected, those measurements were often performed by couples or trios of volunteers. This explains the total number of volunteers specified for Zaragoza, i.e., 233. Figure 9 illustrates the age distribution of the participants, showing that younger individuals were predominant in the Zaragoza pilot, middle-aged users were predominant in Maroussi, and older participants in Ancona. Additionally, due to the fact that complete classes of secondary education students were involved in Zaragoza, this pilot had a significantly higher number of participants compared to the others (Figure 15a,b).

Regarding the role of citizens (Figure 16a), the “working bee” is the predominant role, as intended, since it represents the main type of members of the community of practice who collect and reflect upon the collected data, while the other roles are less represented. In terms of education level (Figure 16b), Maroussi had a higher proportion of citizens with a bachelor’s or master’s degree. In Zaragoza, most participants had a secondary education or a high school diploma. In Ancona, the majority held either a master’s degree or a high school diploma.

Figure 17a illustrates the citizens’ proficiency levels with digital tools. In Zaragoza, most participants had an intermediate or advanced level of proficiency, since they were digital native youngsters. In Maroussi, the majority possessed an advanced level, as they were middle aged people, while in Ancona, the users were evenly divided between advanced and intermediate proficiency levels. They were elderly people with a middle–high educational level.

Regarding work status (see Figure 17b), most of the volunteers in Zaragoza were students, while in Maroussi, the majority were employed people. In Ancona, most participants were retired, which aligns with the demographic analysis of the age distribution of the users.

This section discusses the KPIs that were designed to evaluate the extent to which SOCIO-BEE democratizes CS experiments, particularly those related to air pollution. To assess whether SOCIO-BEE enhances the co-design and co-delivery of evidence-gathering CS experiments that support decision-making in air quality, data were collected from tool logs, databases, and responses from a set of Google Forms surveys created for the project. Below, the most significant KPIs used to inform evaluation dimensions are analyzed.

Only responses from users who completed both the pre- and post-surveys are considered; therefore, 7 citizens in Ancona, 50 in Maroussi, and 103 in Zaragoza were included in this analysis.

The KPIs can be divided into two types: (1) those that measure an increase in favorable user perception towards Citizen Science by comparing pre- and post-intervention dimensions, and (2) those evaluated solely based on post-intervention data. The first category includes metrics such as the increase in technology acceptance, heightened interest or engagement in science, the percentage of EU citizens who feel more aware of air pollution issues after participating in a SOCIO-BEE project, and user behavior changes driven by SOCIO-BEE. The second category includes KPIs that assess aspects such as user satisfaction with SOCIO-BEE (AcadeMe), tool accessibility, inclusiveness, and perceived usability. All values are calculated using the mean and standard deviation (SD) of user responses to the survey questions. Responses were measured on a Likert scale from 1 to 5, where 1 represents ‘strongly disagree’ and 5 indicates ‘strongly agree’.

In the case of the first category of KPIs, the means and SD for both the pre- and post-intervention questionnaire answers, as well as the increase in mean values, were calculated. These values are shown in Table 10, Table 11 and Table 12, for the Ancona, Maroussi, and Zaragoza pilots, respectively.

In the Ancona pilot, some results (acceptance and air pollution awareness) showed a decline in the post-intervention values. This can be attributed to the average age of participants who had limited technological literacy and found the technology that they were offered not entirely usable. Consequently, the decrease in the technology acceptance rate KPI is understandable. Similarly, the fact that understanding the importance of air quality and other factors is not entirely straightforward may also explain the reduction in the percentage of citizens who felt an increased awareness and knowledge about air pollution. The respondents learned about air pollution, but they also realized that there is still a lot more to learn on this matter. Notably, all the average values obtained for the indicators considered in the case of Ancona were higher than 3.69 over 5.0, i.e., over 73% positive perception, which can be regarded as positive, since volunteers felt that their acceptance rate for SOCIO-BEE’s technology and approach, their engagement in CS, and their knowledge about air pollution had increased and their behavior had changed positively.

The Maroussi pilot demonstrated a positive increase in all measured KPIs. Since this pilot involved middle-aged participants with a high level of technological acquaintance and high educational level, they were able to comprehend concepts about air quality and use the technology efficiently. Therefore, the values obtained indicate their positive opinion about SOCIO-BEE in terms of its technology and approach, how engagement is promoted by the project, their awareness about air quality, and their positive influence regarding practices that limit air pollution.

In the Zaragoza pilot, discrepancies were observed in the post-intervention results, with all examined KPIs showing a decrease, exhibiting the biggest decrease of 0.33 out of 5 points. This could be attributed to the involvement of younger participants who are experienced with modern mobile apps and are very sensitive to the usability and user experience of the apps they use on a daily basis. Other factors such as the short duration of the campaigns and technical issues encountered during the interventions could have also affected these results. Remember that the campaigns in Zaragoza were conducted in schools, and each session lasted no more than an hour. Therefore, comparing pre- and post-intervention evaluations in this case may not be highly significant, given the limited interaction period of the citizens with the system. For the second category of KPIs, comprising satisfaction, accessibility, inclusiveness, and usability, the mean and standard deviation were also calculated, but in this case, they were calculated exclusively based on the post-survey responses. The results for each measured value are presented in Table 13. All these values confirm that the measured factors—acceptability, satisfaction, inclusion, and usability—were positively rated across the three population groups corresponding to the three pilots. These findings support the core hypothesis of this paper, demonstrating that the SOCIO-BEE technology and its approach to organizing, executing, and evaluating citizen science (CS) campaigns effectively meet key criteria that are necessary for widespread adoption. Specifically, the solution achieved an average satisfaction agreement of over 70%, accessibility of over 76%, inclusiveness of over 73%, and usability above 60%. These strong results across fundamental evaluation dimensions indicate that SOCIO-BEE is making significant progress toward the democratization of CS campaigns. The consistently high levels of agreement across these multiple evaluation dimensions affirm the platform’s potential to engage a broad and diverse range of participants in meaningful scientific contribution.

### 4.4. Citizen Science and Policy Integration: Lessons from SOCIO-BEE

One of the key challenges in citizen science (CS) initiatives is ensuring that citizen-generated data are not only collected but also meaningfully integrated into policymaking and urban planning. SOCIO-BEE has addressed this challenge by implementing a structured participatory framework across three European pilot cities—Ancona, Maroussi, and Zaragoza. The project demonstrated that CS could bridge the gap between community engagement and policy impact by aligning citizen-collected air quality data with municipal decision-making processes.

The Ancona pilot exemplified how CS can inform targeted public health and urban mobility policies. The project engaged senior citizens in air quality monitoring, revealing pollution hotspots in high-traffic areas like the city center and port. The findings underscored the need for pedestrian-friendly zones and alternative traffic routes to minimize exposure to harmful pollutants. The Municipality of Ancona recognized the value of these data and proposed the integration of CS-driven insights into future urban planning and social assistance strategies.

In Maroussi, the project demonstrated the feasibility of integrating citizen-generated air quality data into municipal monitoring frameworks. By engaging commuters and deploying wearable sensors on public transport, SOCIO-BEE provided real-time pollution data that complemented traditional monitoring stations. The municipal government actively participated in data validation efforts and acknowledged the potential for CS to enhance evidence-based traffic management policies. Additionally, high-profile local policymakers supported the initiative, strengthening its legitimacy and potential for policy uptake.

The Zaragoza pilot showcased the long-term potential of CS by embedding air quality monitoring into local education systems. Engaging students in scientific data collection not only raised environmental awareness but also positioned CS as a continuous, scalable policy tool. The Municipality of Zaragoza integrated the project’s findings into its broader climate neutrality strategy and encouraged further citizen participation in air quality governance.

The results of the SOCIO-BEE projects highlight that the successful policy integration of CS data depends on three critical factors:

Validation and standardization—ensuring that citizen-collected data meet scientific and regulatory standards to be considered in official decision-making;Institutional collaboration—establishing formal partnerships between CS initiatives and municipal authorities to create structured pathways for policy adoption;Community engagement—building long-term citizen involvement through education, awareness campaigns, and participatory urban planning.

By addressing these factors, SOCIO-BEE has established a replicable model for embedding citizen-generated environmental data into local and national policymaking. Future efforts should focus on scaling this approach, fostering cross-border collaborations, and further integrating CS methodologies into formal environmental governance structures.

## 5. Conclusions

The analysis of the validity of the data obtained through the WSNs developed in this paper is presented to ensure the air quality of the studied cities. This research, created in the context of SOCIO-BEE project, successfully demonstrates the potential of citizen science as an effective tool for air quality monitoring and community engagement. By integrating advanced technological solutions with a participatory framework, SOCIO-BEE overcomes key challenges in CS, including data quality assurance, sustained citizen involvement, and policy integration. The project’s campaigns across Ancona, Maroussi, and Zaragoza validate that CS initiatives can produce reliable environmental data while fostering public awareness and action.

Future work will focus on scaling up SOCIO-BEE’s approach to additional cities and refining its methodology to enhance its long-term engagement and impact. Further research will also explore the integration of AI-driven analytics to improve data validation processes and extend the project’s applicability to other environmental challenges beyond air quality monitoring.

## Figures and Tables

**Figure 1 sensors-25-03739-f001:**
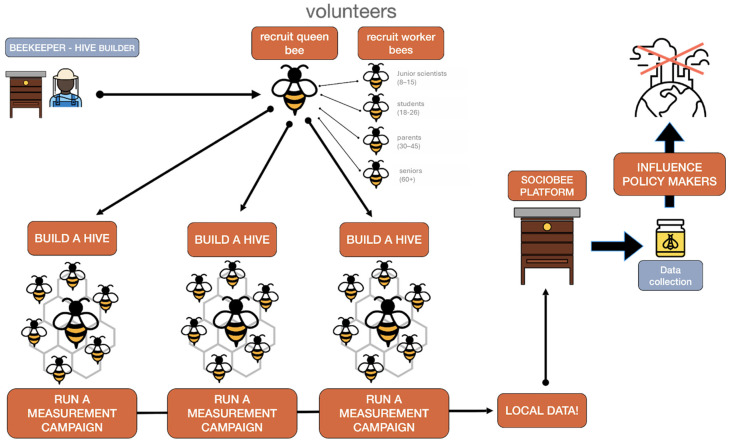
Bee metaphor employed by SOCIO-BEE and co-creation process carried out by a hive [16].

**Figure 2 sensors-25-03739-f002:**
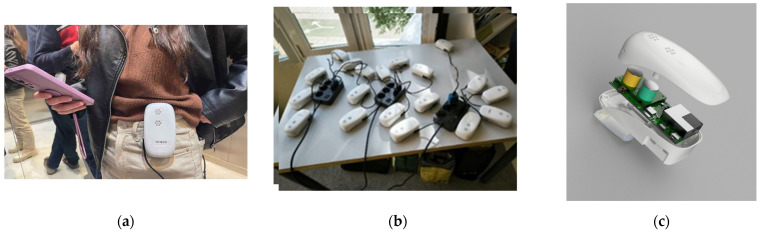
Wearable sensors designed for SOCIO-BEE, ready for use in a CS campaign. (**a**) Example of how to wear the WSN. (**b**) Fleet of WSNs developed for the project. (**c**) WSN design and case.

**Figure 3 sensors-25-03739-f003:**
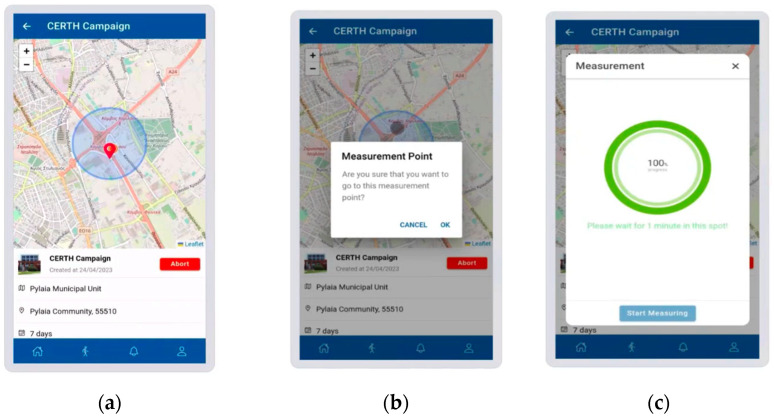
Mobile app showing how to (**a**) select a campaign, (**b**) choose an air quality measurement recommendation, and (**c**) take an actual measurement.

**Figure 4 sensors-25-03739-f004:**
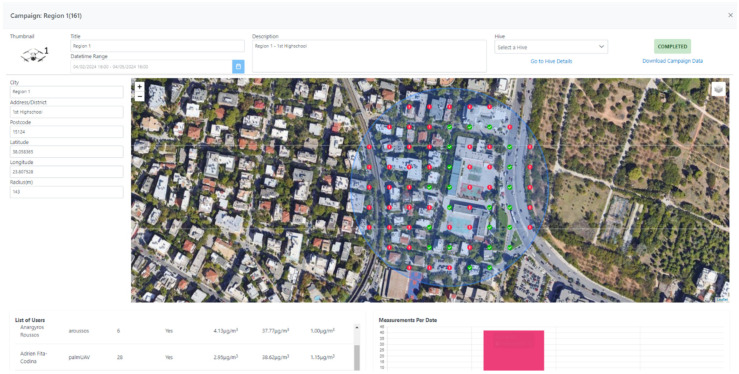
Web app used for designing a citizen science campaign.

**Figure 5 sensors-25-03739-f005:**
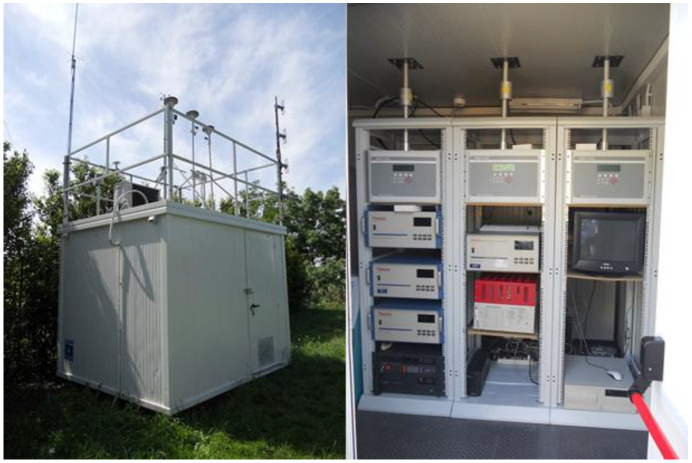
Pictures of the reference station in the city of Ancona.

**Figure 6 sensors-25-03739-f006:**
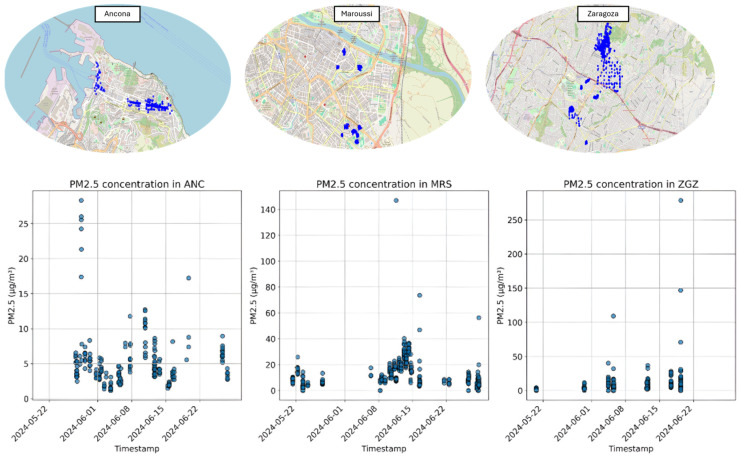
Upper panel: spatial visualization of the area of the city monitored through SOCIO-BEE initiatives. Lower panel: raw PM_2.5_ data collected from the CS experimental campaigns.

**Figure 7 sensors-25-03739-f007:**
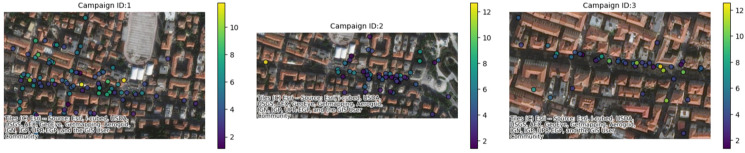
Example of where the measurements were conducted for some campaigns in the city of Ancona.

**Figure 8 sensors-25-03739-f008:**
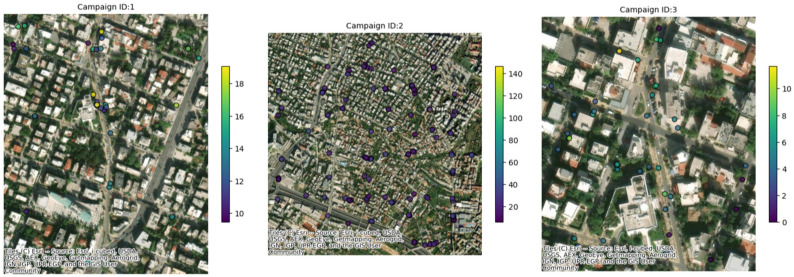
Example of where the measurements were conducted for some campaigns in the city of Maroussi.

**Figure 9 sensors-25-03739-f009:**
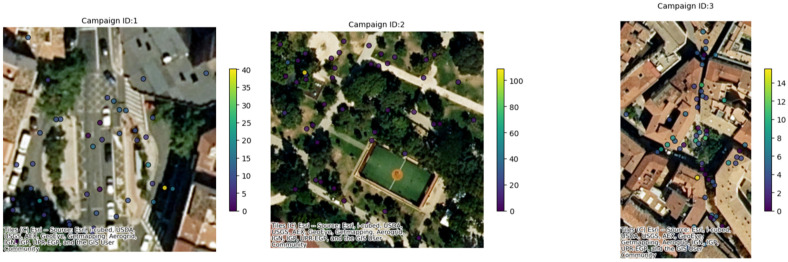
Example of where the measurements were conducted for some campaigns in the city of Zaragoza.

**Figure 10 sensors-25-03739-f010:**
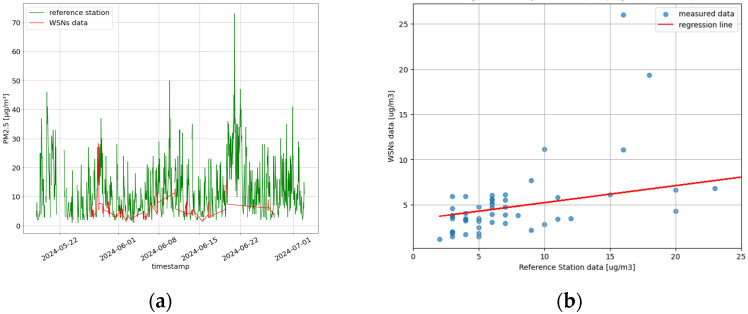
(**a**) Temporal series of the reference station in Ancona and the measurement conducted during the CS campaigns. (**b**) Regression analysis.

**Figure 11 sensors-25-03739-f011:**
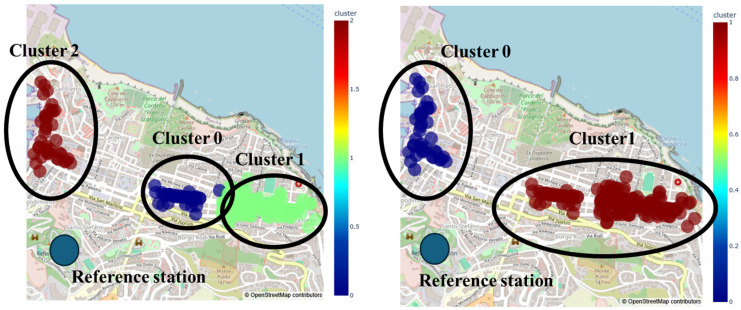
Example of how the clustering algorithm groups the PM_2.5_ data (K = 3, **left** and K = 2, **right**).

**Figure 12 sensors-25-03739-f012:**
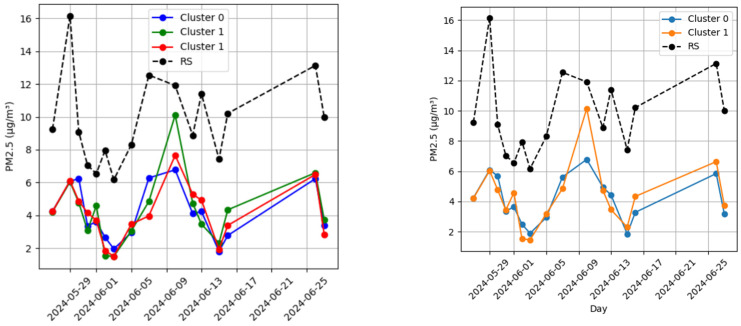
Average daily concentration of PM_2.5_ measured using WSN during experimental campaign, according to the cluster grouping (K = 2, **left** and K = 3, **right**) and measured through reference stations (RSs).

**Figure 13 sensors-25-03739-f013:**
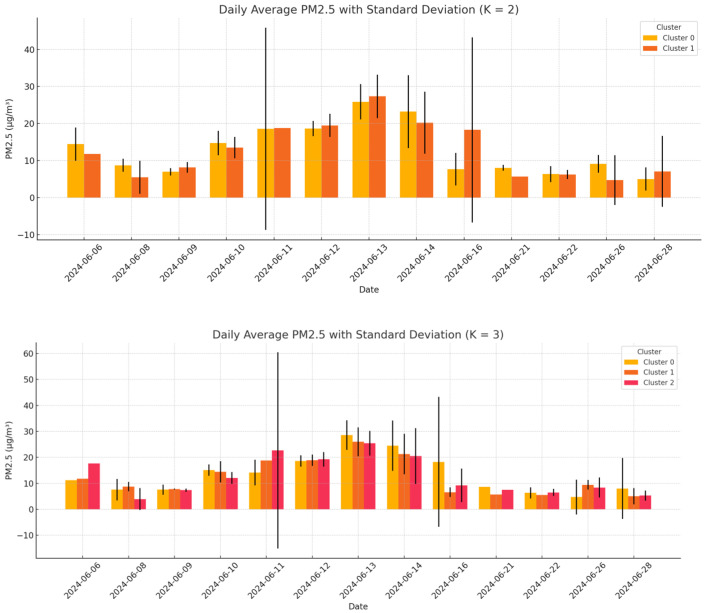
Distribution for each day of monitoring of the average PM_2.5_ for each identified cluster and the related dispersion (Maroussi).

**Figure 14 sensors-25-03739-f014:**
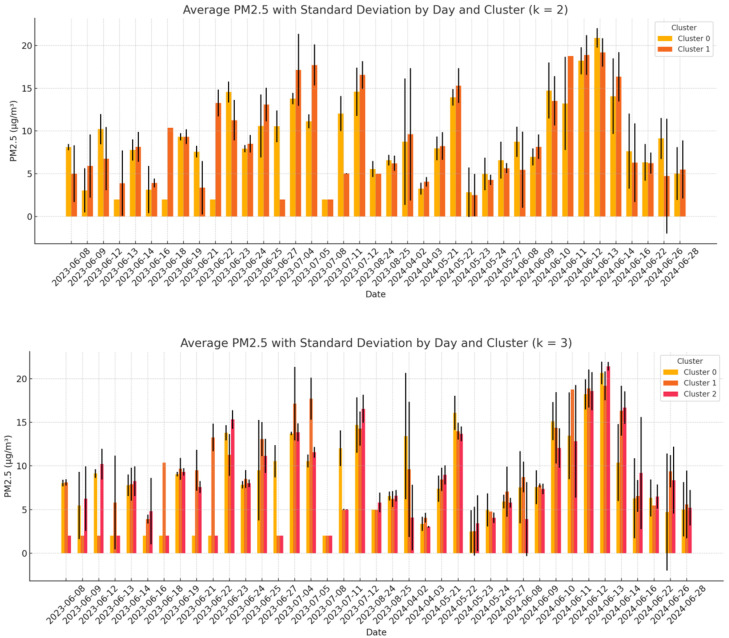
Distribution for each day of monitoring of the average PM_2.5_ for each identified cluster and the related dispersion (Zaragoza).

**Figure 15 sensors-25-03739-f015:**
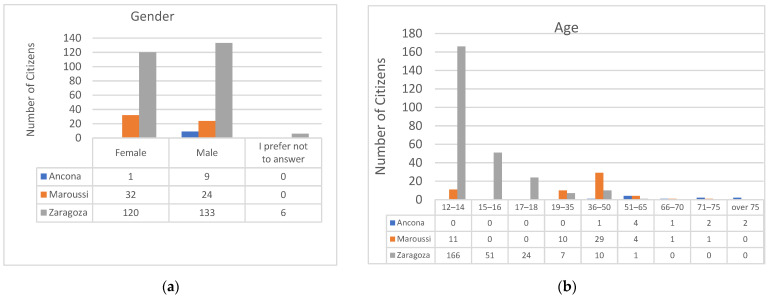
(**a**) Gender analysis and (**b**) age distribution of citizens in pilots.

**Figure 16 sensors-25-03739-f016:**
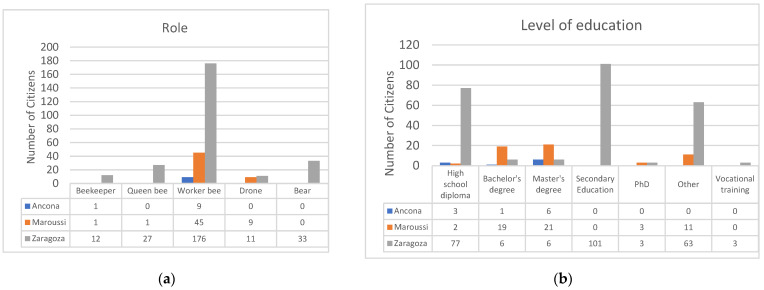
(**a**) Role of citizens and (**b**) level of education.

**Figure 17 sensors-25-03739-f017:**
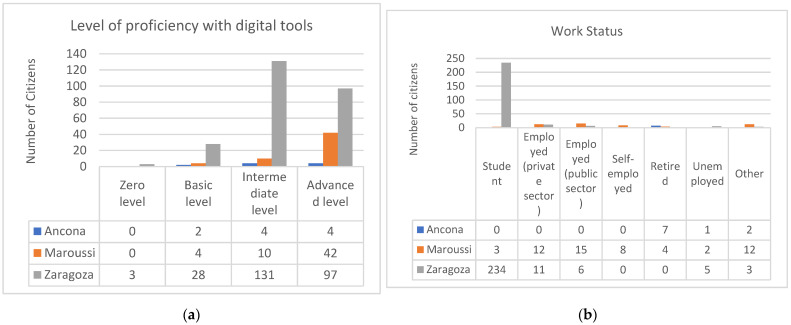
(**a**) Digital proficiency level and (**b**) work status of citizens in pilots.

**Table 1 sensors-25-03739-t001:** Characteristics of the measurement campaigns conducted in the pilot cities.

	Ancona	Maroussi	Zaragoza
Campaigns	6	34	13
Citizens	10	59	79
Overall duration [measurements’ days]	31	30	29
Average daily number of measurements [µ ± σ]	12 ± 7	57 ± 24	24 ± 22

**Table 2 sensors-25-03739-t002:** PM_2.5_ data from Ancona (IT), collected by elderly people.

Campaign	Duration (Days)	Measurement Points	Citizens	PM_2.5_ (µ ± σ), µg/m^3^		
1	32	86	6	5	±	2.2
2	32	55	7	4.2	±	2.1
3	18	44	5	5.1	±	2.7
4	23	23	5	5.4	±	4.1
5	9	7	3	15.3	±	12.1
6	22	10	5	6.3	±	7.0

**Table 3 sensors-25-03739-t003:** PM_2.5_ data from Maroussi (GR) collected by commuters.

Campaign	Duration (Days)	Measurement Point	Citizens	PM2.5 (µ ± σ), µg/m^3^		
1	2	45	8	11.5	±	4.2
2	5	97	10	3.5	±	2.9
3	4	42	3	9.5	±	5.3
4	5	139	1	21.3	±	13.3
6	1	26	8	11.3	±	15.5
7	1	10	2	6.6	±	1.5
8	3	39	6	4.8	±	2.7
9	3	37	12	7.0	±	4.2
10	1	34	4	6.9	±	9.3

**Table 4 sensors-25-03739-t004:** Single-day PM_2.5_ measurements in Zaragoza, Spain, involving schoolchildren.

Campaign	Duration (Days)	Measurement Point	Citizens	PM_2.5_ (µ ± σ), µg/m^3^		
1	1	27	7	1.4	±	1.1
2	1	8	4	3.7	±	1.1
3	1	25	10	3.8	±	2.8
4	1	46	13	10.0	±	6.2
5	1	46	14	7.9	±	16.2
6	1	56	15	4.3	±	3.2
7	1	19	6	6.3	±	3.7
8	1	23	9	10.5	±	9.6
9	1	29	12	4.9	±	3.5
10	1	29	8	10.0	±	7.1
11	1	40	7	8.7	±	3.8
12	1	28	8	7.9	±	6.2
13	1	29	7	25.0	±	56.7

**Table 5 sensors-25-03739-t005:** Results of the correlation analysis between the reference station and the CS campaigns.

Accuracy (µg/m^3^)	Bias (µg/m^3^)	Precision (µg/m^3^)	Statistical Confidence (K = 2) (µg/m^3^)
0.54	3.9	2.2	3.3

**Table 6 sensors-25-03739-t006:** Summary of the comparison between clustered citizen science (CS) data and the reference station (Ancona). The table reports the mean percentage error, standard deviation, and Pearson correlation coefficient for each cluster under two clustering configurations (K = 2 and K = 3), providing insight into the accuracy and consistency of the CS measurements.

Cluster	Mean Relative Error [%]	Standard Deviation [%]	Pearson Correlation with the RS
K = 2			
0	55.05	16.6	0.75
1	56.92	13.37	0.63
K = 3			
0	55.59	18.2	0.74
1	57.88	11.5	0.63
2	56.07	13.3	0.68

**Table 7 sensors-25-03739-t007:** Pearson correlation between the distance of cluster centroids from the reference station (Ancona) and the corresponding percentage error in PM_2.5_ measurements for different clustering configurations (K = 2 and K = 3).

Cluster	Pearson
2	−0.02
3	+0.03

**Table 8 sensors-25-03739-t008:** Standard deviation of the PM_2.5_ for each identified cluster and the related coefficient of variation (Maroussi).

	Standard Deviation [µg/m^3^]	Coeff of Variation
K = 2	5.9 ± 6.9	0.5 ± 0.4
K = 3	5.8 ± 7.5	0.5 ± 0.4

**Table 9 sensors-25-03739-t009:** Standard deviation of the PM_2.5_ for each identified cluster and the related coefficient of variation.

	Standard Deviation [µg/m^3^]	Coeff of Variation
K = 2	2.3 ± 1.6	0.3 ± 0.3
K = 3	2.1 ± 1.7	0.28 ± 0.3

**Table 10 sensors-25-03739-t010:** Ancona KPIs.

Ancona (µ ± σ)			
KPIs	Pre	Post	Increase (Average)
Technology acceptance rate	4.06 ± 0.80	3.69 ± 1.00	−0.37 ± 0.21
Higher interest or engagement in science	3.66 ± 0.92	3.81 ± 0.98	0.13 ± 0.05
Percentage of EU citizens who feel more aware of air pollution issues after participating	4.36 ± 0.51	4.14 ± 0.69	−0.21 ± 0.18
User behaviour changes driven by the project.	3.67 ± 0.93	4.05 ± 0.78	0.38 ± (−0.15)

**Table 11 sensors-25-03739-t011:** Maroussi KPIs.

Maroussi (µ ± σ)			
KPIs	Pre	Post	Increase (Average)
Technology acceptance rate	4.14 ± 0.76	4.42 ± 0.76	0.28 ± 0.00
Higher interest or engagement in science	3.67 ± 0.92	3.83 ± 0.88	0.17 ± (−0.04)
Percentage of EU citizens who feel more aware of air pollution issues after participating	4.06 ± 0.79	4.40 ± 0.81	0.34 ± 0.02
User behaviour changes driven by the project.	3.38 ± 1.02	4.04 ± 0.78	0.66 ± (−0.25)

**Table 12 sensors-25-03739-t012:** Zaragoza KPIs.

Zaragoza (µ ± σ)			
KPIs	Pre	Post	Increase (Average)
Technology acceptance rate	4.20 ± 0.80	4.20 ± 0.80	−0.32 ± 0.23
Higher interest or engagement in science	3.72 ± 1.04	3.58 ± 1.00	−017 ± (−0.04)
Percentage of EU citizens who feel more aware of air pollution issues after participating	4.19 ± 0.71	3.86 ± 0.99	−0.33 ± 0.27
User behaviour changes driven by the project.	3.75 ± 1.02	3.66 ± 0.98	−0.09 ± (−0.04)

**Table 13 sensors-25-03739-t013:** Cross-pilot satisfaction, accessibility, inclusiveness, and usability KPIs.

Mean ± SD			
KPIs	Ancona	Maroussi	Zaragoza
User appreciation (satisfaction) with SOCIO-BEE (AcadeMe),	3.36 ± 1.43	4.26 ± 0.87	3.97 ± 1.00
The level of tool accessibility	4.29 ± 0.74	3.81 ± 1.08	3.92 ± 0.97
The level of inclusiveness of the tool	3.86 ± 0.9	3.65 ± 0.99	3.73 ± 0.96
Perceived usability score	3.09 ± 1.24	3.14 ± 0.87	3.51 ± 1.09

## Data Availability

Data is contained within the article.

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
