# Peer review of "A Wearable Sensor Node for Measuring Air Quality Through Citizen Science Approach: Insights from the SOCIO-BEE Project"

_sensors, 2025, doi:10.3390/s25123739_

Round 1
Reviewer 1 Report
Comments and Suggestions for Authors
It is important to utilize Citizen Science's power to enhance data sampling in environmental science research.
However, I believe there are still a few published examples in academic literature with Citizen Science in the field of air quality monitoring.
This paper reports on the implementation of air quality monitoring, and I consider it valuable that the data be published in a journal.
However, the authors need to revisit the paper regarding the following points:
1) Line 191
NO2: ”2” should be written as a subscript.
2) Line 224
PM2.5: Only decimal points are not subscribed.
PM2.5 and PM10: Numbers for PMs are not subscripted in other sentences. Authors should be consistent in their formatting.
3) Table 8: Do not use highlighters.
4) Line 805: Table “9”; Do not use Italic.
5) Line 902: Write the number of the Table. (Table 13)
6) Table 13: Coeff of variation; Coefficient of variation
7) Line 917: Table “4” should be “14”
8) Line 996: extend the project’s applicability
Change the sentence to “extend the applicability of the project”
Reviewer 2 Report
Comments and Suggestions for Authors
The paper “Leveraging Citizen Science to Gain Reliable Data Evidence on Air Pollution: Insights from the SOCIO-BEE Project" describes several aspects of the SOCIO-BEE concept realized through the project of the same name. Citizen science engages the public in various activities that can provide new advantages for scientific research and policymakers, benefiting society as a whole. The SOCIO-BEE project is designed to cover all aspects of the infrastructure for collecting and post-processing air pollution data by individuals or groups. This concept has been tested, and the paper illustrates performance. Its publication may help others create and implement similar ideas to pursue analogous approaches. In this sense, it is very difficult to review something that already functions as such. However, from a scientific perspective, it would be beneficial to conduct a more thorough review of the sensors used and the results obtained, specifically regarding their applicability in scientific studies. The only complaint pertains to this segment of the work.
Author Response
Please see attachment-

Reviewer 3 Report
Comments and Suggestions for Authors
Thanks to the authors for such a large manuscript. I read it with pleasure - I remembered my youth, when the social mechanisms described in the manuscript were discussed with such companies as Nokia, Samsung, Siemens, etc.! In many ways, the idea of integrating a gas sensor into a mobile phone did not work out due to the social aspects partly discussed in the manuscript.
But....The official aims of the Sensors MDPI journals is -https://www.mdpi.com/journal/sensors/about
Based on aims of the Sensors MDPI journal I have several technical questions and recommendations!
- The current version of the manuscript does not cover technical aspects of using gas sensors (electrochemical sensors shown in Figure 3)...Without technical aspects of using gas sensors, but only social ones, the manuscript is more suitable for the journal -https://www.mdpi.com/journal/sustainability
- Surprisingly, the research process selected pair highly problem pollutants of the air atmosphere (line 292) - key air pollutants (𝑂3, 𝑁𝑂2, and 𝑃𝑀2,5). NO2 and O3 are have big cross sensetivity in to electrochemical sensors (please look page 4 in data sheet https://www.membrapor.ch/sheet/Ozone-Gas-Sensor-O3-M-100.pdf )
- It is important for you to add a section with a description of used Wireless Sensor Nodes (WSNs) which include direct links to data sheets of electrochemical sensors!!! After this, one can judge about nature of the measured data and compare them with the basic state climate stations located in the cities.
Reviewer 4 Report
Comments and Suggestions for Authors
The paper <Leveraging Citizen Science to gain reliable data evidence on air pollution: insights from the SOCIO-BEE project> contains interesting information about the use of standard sensors to monitor the state of the atmosphere in cities. These sensors are worn by volunteers and transmit information to the center for further processing. The SOCIO-BEE project is of interest, and it may have advantages over conventional methods of monitoring the atmosphere based on the use of stationary devices. However, there are disadvantages that are not considered in this paper. By itself, a human volunteer can also be considered as a source of atmospheric pollution, his metabolism can have a significant impact on the sensor readings. The paper focuses on the application of sensors, but not on the sensors themselves. Meanwhile, the journal is dedicated to sensors. It is difficult to say how much this paper corresponds to the problems of the Sensors | Topical Advisory Panel magazine.
In my opinion, this paper would be more suitable for Atmosphere journal, or it should be supplemented with information about new developments in the field of sensors and the influence of the human volunteer body on sensor response.
Round 2
Reviewer 3 Report
Comments and Suggestions for Authors
Thanks to the authors for the great work done to improve the manuscript. As I understand it is impossible to provide more info about sensor node than what is presented on the website https://bettaircities.com/bettair-node/
I have no more questions about sensory technologies.
The manuscript currently requires editing. There are not all references are correctly reflected in the text!
Good luck to the authors in their further research!
Reviewer 4 Report
Comments and Suggestions for Authors
The answer to the previously posed question is missing:
What influence does the human body have on the sensor's response?
